# 1 Feasibility of small wind turbines in Ontario: Integrating power

#### 2 curves with wind trends.

- 3 Masaō Ashtine<sup>1</sup>, Richard Bello<sup>1</sup> and Kaz Higuchi<sup>1</sup>
- 4 <sup>1</sup> York University, 4700 Keele Street, Toronto, Ontario, Canada.
- 5 Correspondence to: Masaō I. Ashtine (mia35@cam.ac.uk)
- 6

#### 7 Abstract

8 Micro-scale/small wind turbines, unlike larger utility-scale turbines, produce 9 electricity at a rate of 300 W to 10 kW at their rated wind speed and are typically 10 below 30 m in hub-height. These wind turbines have much more flexibility in their 11 costs, maintenance and siting owing to their size and can provided wind energy in 12 areas much less suited for direct supply to the grid system. The small wind industry 13 has been substantially slow to progress in Ontario, Canada, and there is much debate 14 over their viability in a growing energy dependent economy. In an effort to diversify 15 the energy sector in Canada, it is crucial that some preliminary research be conducted 16 in regards to the relevance of changing winds as they impact small wind turbines; this 17 study seeks to demonstrate the performance of two small wind turbines, and speculate 18 on the potential power output and its trend over Ontario historically over the last 33 19 years using the North American Regional Reanalysis (NARR) data. We assessed the 20 efficiencies of a Skystream 3.7 (2.4 kW) and a Bergey Excel 1 kW wind turbines at 21 the pre-established Kortright Centre for Conservation wind test site, located north of 22 Toronto. We have found that the small turbine-based wind power around the Great 23 Lakes and eastern James Bay have increased during the seasonal months of winter 24 and fall, contributing as much as about 10% in some regions to the total electricity 25 demand in Ontario.

# **KEYWORDS**

small wind turbines; NARR dataset; multi-year wind trends; wind turbine power curves;

renewable energies

# 5 Correspondence

Masaō I. Ashtine -Present Address- Department of Geography, Centre of Atmospheric Science,

University of Cambridge, Cambridge CB2 1TN, United Kingdom.

E-mail: mia35@cam.ac.uk

Tel: (44) 0758 2597896

10

## **1** Introduction

- Much of the modern wind turbines representing the renewable energy landscape consist of large utility-scale wind turbines which can produce electricity in the magnitudes of MW (megawatts), taking advantage of stronger winds aloft with higher hub-heights and larger rotor diameters. Canadian investment in wind turbines saw a 20% growth in clean wind energy production in 2012, representing over \$2.5 billion in investment, and Canada's current installed capacity is just
- over 6.5 GW (gigawatts).<sup>1</sup> The province of Ontario has 2.9 GW of installed wind capacity, ~ 30% of Canada's total capacity.<sup>2</sup> The small wind turbine industry however is focused on the installation of wind turbines which produce electricity on an average between 300 W to 10 kW (kilowatts) rated power with hub-heights that are generally below 30 m. Although small-scale wind turbines have been around historically, employed for different functions like grinding
- grains, they have failed to dominate the wind energy sector owing to increasing doubts about their performance, technological advancements, field-testing and feasibility in a changing climate.

Small wind turbines produce more costly electricity than their utility-scale counterparts, especially in poor wind sites. When tailored to specific wind regimes, and used at optimal

- conditions through wind site assessment, small wind turbines can be a reliable energy source and socio-economic benefit to regions disconnected from the grid. Seen as a means to increasing electrical supply to small isolated communities in developing countries<sup>3</sup>, the small wind industry has been especially hindered in Canada with currently between 2,200 and 2,500 turbines installed, 90% of which fall into the 'mini' wind turbine category (< 1 kW rated power). The</p>
- total combined capacity of all SWTs (Siemens Wind Turbines) is estimated to be between 1.8

MW and 4.5 MW, equivalent to the capacity of one to three modern utility-grade wind turbines, with annual output roughly at 7.5 GWh (gigawatt hour) per year, equivalent to an amount of electricity consumed by approximately 750 Canadian homes.<sup>1</sup>

The small wind industry has afforded the renewable energy sector with the benefits of energy

- independence for the consumer, remote electricity production in regions off-grid and a more diversified energy supply which can be complemented with solar energy and utilized by businesses and households. However, this industry is faced with many challenges, particularly the lack of standardized field testing of these wind turbines, resulting in uncertainty in performance claims by manufacturers. A vast amount of testing done to establish small wind
- turbine rated power and power curves is done in wind tunnels, focusing on the electrical components of the wind turbines and not realistically assessing turbine performance in the field. As environmental factors such as temperature, radiation and wind variability affect turbine performance, field-testing is essential. Studies assessing the performance of small wind turbines in the field have often focused on the turbine's effect on the local environment or turbulence
- <sup>60</sup> patterns produced by secondary rotor effects.<sup>4-9</sup> Since there are currently no formal standardized testing regulations for their calibration and power output in the North American wind industry<sup>10</sup>, it is difficult to develop a small-scale diversified electrical generation strategy under a changing wind field caused by global warming. Our study is a first step to address this issue in Ontario.
- 65 The Kortright Centre for Conservation has been at the forefront of renewable energy initiatives in Toronto, Ontario, being one of two main test sites for standardization of small wind turbines in Canada (second is located in Prince Edward Island). Two leading industry standard turbines were assessed in this study, the Bergey Excel 1 kW and the Skystream 3.7 (2.4 kW) wind turbines

with hub-heights of 16.8 m and 15.2 m, respectively. These turbines have varying specifications,

- as listed in Table 1. Our study seeks to understand the historical (33 year, 1980 2012) electrical output potential for small wind turbines using the NARR wind data at 10 m and 30 m over Ontario. We have incorporated highly optimized wind power curves of our two small wind turbines into the NARR data over the 33-year period, finding trends in the electrical output that best demonstrate the potential of this industry, spatially and temporally across Ontario.
- 75

## 1.1 NARR dataset

The NCEP-NARR is a high-resolution atmospheric and land surface hydrology dataset for the North American domain.<sup>11</sup> At present this dataset comprises of reanalysis data for the period

- 1979 present; in the present study, 3-hourly data from 1980 2012 were used. The widely known NARR procedure uses the very high resolution NCEP Eta Model (32 km, 45 layers) together with the Regional Data Assimilation System (RDAS).<sup>12-15</sup> NARR is widely known for its successful assimilation of high-quality and detailed precipitation observations into the atmospheric analysis which was previously lacking from many global models. This research
- focused on the electrical output potential over Ontario for the tropospheric heights of 10 m and 30 m, hub-heights relevant to small wind turbines. However, owing to a preliminary coding error, 43 grid cells along the lower Hudson Bay coastline were found to be incorrect at the 30 m level, due to their low lying elevation

(http://www.emc.ncep.noaa.gov/mmb/rreanl/faq.html#zero-30m-winds). However, these grid

cells (0.01% of the study area) were found to be non-influential on the neighbouring cells and were omitted from the analysis.

## 95 2 Methods

## 2.1 Power curves

The Kortright test site is located short distance (~30 km) north of Toronto, with an open fetch, having a predominantly southeast and northwest wind pattern. Meteorological and electrical

- 100 output data were captured for both turbines between 7 November 2012 and 30 April 2013. These data provided 5-second readings of wind speed at 8 levels above ground, temperature, wind direction and turbine power output for ~ 6 months. Analysis of these data produced performance data through power curve analysis (Fig. 1), demonstrating how the Bergey and Skystream wind turbines performed at differing wind speeds. Applying a best fit curve, 4<sup>th</sup> order polynomial
- equations were obtained for each turbine that best described the ability of the turbine to convert wind energy into electrical power (P) in Watts, Eq. (1) and (2):

Bergey 1 kW  

$$P = -0.101x^4 + 2.02x^3 - 2.878x^2 - 2.187x + 2.732$$
(1)

Skystream 2.4 kW

$$P = -0.144x^4 + 2.985x^3 - 8.246x^2 + 9.301x - 3.978$$
(2)

where *x* is the output wind speed.

#### 2.2 Applying power curves to NARR

Using the NARR wind components, u and v, wind speed U (ms<sup>-1</sup>) at the tropospheric levels of 10 and 30m was calculated using the standard magnitude formula (Eq. 3):

$$U = \sqrt{u^2 + v^2} \tag{3}$$

Wind speeds at 10 and 30 m heights were derived for every 3-hr measurement from the corresponding NARR wind data (1980 – 2012). Monthly mean wind speeds, 33-yr monthly averages and seasonal means were assessed for winter (DJF), spring (MAM), summer (JJA) and fall (SON).

The historical power generating potential for the Bergey and Skystream wind turbines were then calculated by inputting the NARR wind speed data to Eq. 1 and 2, respectively; however, only the data for the Bergey 1 kW wind turbine is presented as it was found that the Skystream 2.4 kW power curve closely represented the Bergey's due to its underperformance. The Bergey

turbine power curve demonstrated electrical output for each 3-hr reading from the NARR dataset in megajoules (MJ) and the summed electrical output for each month was averaged based on 33 years of wind speed data. This method was repeated at the 30 m and spatial differences in

performance between the hub-heights of 10 and 30 m show regions where increases in the hubheight have proven more effective than in other regions.

Trend analysis in the electrical output over Ontario and the Great Lakes from each wind turbine was computed with the OLS (ordinary least squares) method, along with the interannual variability of wind power. Plots of significant trends using *t*-test analysis are reported on a seasonal basis.

## **3** Results and discussion

## 3.1 Seasonal variations in turbine power output

- Seasonal variations and long-term trends in the NARR wind field from 1980 to 2012 at the 10 and 30 m heights have been described in Ashtine et al. (2016). The spatio-temporal patterns of the turbine energy output correspond well with changes in winds discussed in Ashtine et al. (2016). Power curve produced for the Bergey 1 kW (Fig. 1) wind turbine was in close agreement with the power curves of this turbine from the field testing.<sup>16, 17</sup> The Bergey reaches its maximum
- power output of 1.1 kW at 13.5 ms<sup>-1</sup> with a cut-in wind speed of 2.5 ms<sup>-1</sup>. Turbine power output closely follows patterns in mean wind speeds, with the Great Lakes and James Bay producing the greatest amount of electrical energy for both turbines during the winter and fall seasons (Fig. 2). Significant seasonal variations in the power output are observed over the major water bodies, as a result of the impact on surface winds from melting and formation of ice over water.<sup>18</sup>