# Peer review of "Feasibility of small wind turbines in Ontario: Integrating power"

_Wind Energy Science, 2016_

## Referee Comment (RC1) · Anonymous Referee #1 · 19 May 2016

GENERAL COMMENTS:

This study takes an elegant approach to an important issue, the development of electricity production from small wind turbines. Through conducting a field test for two types of turbines and then applying the resulting findings across a map of Ontario's wind resources, this article makes some persuasive conclusions about what regions can most benefit from small turbines and what kinds of turbines are most likely to be productive (i.e. those with higher hub-heights and higher rated outputs). In my opinion, the findings certainly have sufficient scientific significance to merit publication. However, the way in which they are presented must be improved, and there are a number of questions about the methods and data interpretation that should be answered.

One overall problem is the organization of the sections, or perhaps the general research approach just isn't communicated clearly enough. Neither the abstract nor the introduction gives a particularly clear outline of the article or a simple summary of the study (e.g. "First we calculated how much electricity two types of small turbines could produce at different wind speeds, and then we examined provincial wind speed data to determine the overall potential of these turbines in different regions"). I didn't really understand the basic method until about halfway through the article – a lot of emphasis was put on the NARR dataset, which I couldn't appreciate the relevance of until after the section about calculating power curves. I also had trouble following the "results and discussion" section.

Another overall problem is the use of jargon. It's fine to use technical language, but each term needs to be accompanied by a brief explanation when first introduced. If a term is just dropped into the text (e.g. power curve, rated power, NCEP, ordinary least squares) without explanation, then it's jargon and it reduces the article's accessibility. It may also be possible to remove some technical terms from the text completely if they aren't really necessary. Let me say that I am not an engineer or a geographer, so it's possible that the target audience would have an easier time deciphering the article than I did, but I think the authors should want it to be accessible as possible anyway.

Next, the results section focuses too much on the specific details of the analysis, and not enough on the big picture. There needs to be more "so what" statements throughout and more discussion of the implications for policy and investment. Also, this is an international journal, so lessons that would apply outside of Ontario need to be clear, which is all the more reason to synthesize the information and draw general conclusions. This is particularly appropriate since the structure uses a "results and discussion" section rather than a separate section for each of results and discussion. I think the combined structure is fine, but discussion is currently under-emphasized.

I also notice that the literature in the references list is quite technical; there are very few pieces on the broader challenges of wind energy planning. It's possible that adding a few more socially-oriented pieces would help increase the accessibility of the article

and contextualize the generalizability of the findings. I'm not entirely convinced that there needs to be a big change in terms of the literature (especially since I may be outside the target audience), but I do see some opportunities there. Below this I've included just a few possible citations I extracted from a similar article on wind energy. I'm not particularly up-to-date with the literature in the field, but these pieces could certainly be a start if the authors want to supplement what they have.

Cowell R (2010). Wind power, landscape and strategic, spatial planning – The construction of 'acceptable locations' in Wales. Land Use Policy 27: 222-232.

Ek K, Persson L, Johansson M, Waldo A, (2013). Location of Swedish wind power – Random or not? A quantitative analysis of differences in installed wind power capacity across Swedish municipalities. Energy Policy 58: 135-141.

Möller B (2010). Spatial analyses of emerging and fading wind energy landscapes in Denmark. Land Use Policy 27: 233-241.

Toke D, Breukers S, Wolsink M (2008). Wind power deployment outcomes: how can we account for the differences? Renew Sust Energ Rev 12: 1129-1147.

Van der Horst D, Toke D (2010). Exploring the landscape of wind farm developments; local area characteristics and planning process outcomes in rural England. Land Use Policy 27: 214-221.

Let me temper my comments by reiterating that, while I have some experience with the academic study of wind energy planning, I am not from the same field as the authors. I do not feel qualified to assess whether important pieces of literature are missing and I am not qualified to provide a credible assessment of whether the methods are appropriate and reproducible (although I was able to make sense of them in the end). Before being accepted for publication, I expect that this article will also be reviewed by a referee or editor with the appropriate background in engineering or geography.

SPECIFIC COMMENTS:

-abstract stops after basic results, needs a concluding sentence on the significance/implications

-is line 28 referring to investment in large-scale turbines or turbines in general? -and what period of time was this investment made over? just 2012? all years?

-what does "rated power" mean on line 33?

-line 41 implies that small turbines cannot be integrated with the grid – is this true?

-how is the mention of developing countries on line 42 relevant to this Canada-centred study?

-on line 45, are all small wind turbines Siemens turbines? if not, more information is needed

-is line 50 referring to the industry in Canada or internationally? -earlier, implied that the industry is weak in Canada, so how are these benefits possible?

-on line 55, what is rated power and what are power curves?

-if it's obvious that field testing must be done, I'm curious about why it isn't done now

-on line 63, "wind field" sounds like jargon (though I can tell what it means from context)

-say more about the Kortright Centre, otherwise it's not clear why it's being mentioned (line 65)

-why these two types of turbines? why not more than two types? why not other types?

-table 1 is interesting, but is all the information necessary? lots of technical terms -could acknowledge that the technical info is an aside by putting the table in an appendix

-why 33 years? is this just looking at wind data? or have small turbines been around that long?

-on line 71, NARR wind dataset needs to be explained before it is referenced -add a

1-sentence description, or this part could be moved after the NARR dataset section

-why 10m and 30m?

-what are "highly optimized wind power curves"? -what does it mean to "incorporate them into the NARR data"?

-the overall approach should be phrased as an objective or a research question

-the introduction should end with an outline of the rest of the article

-on line 78, what is NCEP? this acronym is never defined

-what is "reanalysis data"? (line 79)

-on line 80, is 3-hourly data the highest resolution available? why was it chosen? -and it's just wind data being used? at 10m and 30m? clarify and be specific

-what is the "NARR procedure"? what are "layers"? what is RDAS? why does it matter?

-why does precipitation data matter for this study? (line 83)

-sentence about coding error needs much more explanation, is it your error or an error in NARR? -if it has such a small influence on the study, is it even worth mentioning at this point?

After the NARR section, I'm not necessarily going to list every specific instance of jargon/technical terminology I see. Have someone from outside the discipline give the article a read and identify areas that are especially difficult to understand. Instances of jargon should either be removed or explained.

-it's not clear how equation 3 is relevant, what are "wind components u and v"?

-why is figure 1 only the Bergey? (I see this is explained later, but figure should have a note) -also, why did the Skystream underperform? is the Bergey curve really accurate for it?

-at line 132, I'm finally starting to understand what this research is trying to do

-on line 135, this was done for each region/cell in the data, right? how big are the cells? 32km?

-what is "trend analysis"? what is the "ordinary least squares" method? (line 141)

-on line 154, what turbine? was the Bergey tested in another study? isn't this the field test? -(upon reflection, is this saying that you reproduced the manufacturer's power curve?) -(if so, what is really being contributed here? what are the implications?)

-on line 159, assuming turbines can't be installed over water, why does wind over water matter?

-trends mean how wind speeds have changed over time due to climate change, right? (line 176)

-increasing the hub-height "from 10m to 30m" – be specific (line 189)

-maybe some summary statements (e.g. "basically. . ."), not just the individual details? (line 198)

-at line 202, the shift to kWh as a unit raises some questions: -bigger regions would have greater production, so how big are these regions? -is this assuming they are covered with turbines? how densely? -these are still estimates, right? nothing to do with where turbines actually are? -use of kWh here may be arbitrary and not useful, or I've missed something fundamental -(upon reflection, this was probably kWh per year per turbine, but not clear from the text)

-section 3.2 is very interesting, but is this analysis coming from the results, the literature, both? -also, explicitly state the implications for turbine siting/investment/development

-is the organization right with section 3.3? wasn't output discussed in the last paragraph of 3.1?

-for that matter, wasn't output discussed all throughout section 3.1? -I thought "trends" here would mean variation from year to year, not seasonal variation...

-on line 238, isn't this just repeating the observations from line 190?

-elaborate on line 269, does each turbine make this much money each year through feed-in tariff? -or is this just a general estimate of the value of the electricity they produce? -what about the capital cost of building the turbine to begin with? is that included? -if this is about feed-in tariff, there's tension with earlier remarks re: off-grid consumers

-numbers on line 279 and on seem a bit disappointing, end the section with something positive?

-why do the seasonal trends matter? what do they mean for siting/investment/development? -I see why regional variation and hub height differences matter, but not seasonal variation

-on line 295, can you say that higher output is better after the Skystream "underperformed"?

-what are the assumptions going into the 5-7% estimate on line 298? each house has one turbine?

-can Bergey data be extrapolated to 10kW + 25kW turbines? why weren't they tested? (line 302)

-use in-text citations not footnotes

TECHNICAL COMMENTS:

14: should be "its" not "their"

15: rephrase sentence

18: "potential power output" of what? the two turbines?

20: "turbine" not "turbines"

23: "increases" not "have increased"

25: "The wind turbines that represent the modern renewable energy landscape are generally…"

26: "on the scale of" not "in the magnitudes of"

27: "high" not "higher" and "large" not "larger"

30: use the word "about" or "approximately" instead of "~"

32: "at an average rate of" not "on an average"

39: add the word "however" or "although" to the beginning of this sentence

41: "increase" not "increasing"

49: "provided" less awkward than "afforded"

54: "most" instead of "a vast amount of"

63: use "climate change" instead of "global warming"

67: "the other" not "second"

After the introduction, I'm going to stop listing small stylistic/grammar errors (unless I notice some that are particularly problematic). Basically, the article needs a bit more editing/revision to address stylistic issues throughout. However, the errors are not an enormous shortcoming; they are generally more distracting than confusing.

192: sudden use of "ca." (instead of "~" or "about") is jarring and inconsistent

---

## Author Comment (AC1) · 31 May 2016

Dear Referee,

On behalf of my co-authors, thank you very much for the constructive feedback on our manuscript. We will be sure to address the points and corrections that you have suggested. Perhaps a brief response to better explain some aspects of our paper, I hope the following help to clarify some issues presented:

- As a wind energy science paper, we understand that some of our terms and jargon are not the most used outside of the field, yet, many of these terms are unavoidable to an extent. However, as per your suggestion, we will certainly preface these terms and concepts for an easier read/understanding. These are simple improvements that can be sometimes overlooked, so thank you for highlighting this issue.

- We understand that we did not sell our methods strongly enough in our Introduction and Methodology and thus this too is a good point that we will take on board when editing the revised manuscript. As stated in your feedback, this will give readers a more sound understanding of our paper and what it attends to achieve.

- You are correct on suggesting an improvement on our bigger picture implications and how our findings can be applied outside of Ontario. We will re-work the discussion section to reflect this more clearly. Thank you for your suggested literature and we will use any important information from these appropriately. However, this paper does not delve heavily into the socio-economic issues affecting the wind industry, but simply addresses the feasibility from a wind energy potential perspective. Thus, we will also edit our abstract to make this clear.

We apologize for the delay in our response but thank you once again for your suggestions and review. Our revised manuscript will highlight these changes in detail for your review.

Best wishes

---

## Referee Comment (RC2) · Anonymous Referee #2 · 9 Jun 2016

It is a good paper on investigation of feasibility of small wind turbines in Ontario. Due to the greatest wind speed trend during winter and fall at the hub-heights between 10 and 30 m, small wind turbine (rated of 300 W to 10 kW) is possible to largely implement. A lot of real data shows that even with a 1 kW wind turbine, 130-200 dollars can be saved annually in Ontario.

---

## Referee Comment (RC3) · Anonymous Referee #3 · 17 Jun 2016

General comments:

The power production of two chosen wind turbines installed on a wind field test site (Kortrick site) during 6 month was used to calibrate the power curve in real working configurations. Then, using the dataset from the North American Regional Reanalysis (NARR) over Ontario at two heights (10m and 30m), the production from the chosen wind turbines was estimated over 33 years. The analyses were performed using 33year monthly averages and seasonal means, allowing an historical power generating potential for the chosen wind turbines.

The work is interesting and indeed very important for improving the use of small wind turbines. From what I understood, the originality of the present work is to use on-site calibrations of the wind turbine power curve to adjust the power output to real working

conditions of wind turbines.

I recommend the publication of this interesting work with however a major revision that includes:

- A more focused objective that will lead to a much clearer introduction. I would focus on the winter season for the NARR dataset as it corresponds to the calibrated power curve. Also, I would analyze the NARR dataset with a month by month variations of the calibrated power curve versus the power curve given by manufacturers.

- An additional bibliography on the use of reanalyzed data for power output estimation, using power curves from manufacturers. This will emphasize the originality of your work.

Detailed of major issues:

The introduction is really unclear. You can find three examples bellow:

p3 L61: "no formal standardized testing regulations for their calibration and power regulation". From what I understood, you are introducing your study as a potential way to improve standardization. However, it is not clear on how your study will contribute to this aspect. Can you please develop and clarify more this point ?

P3 L65:"The Kortright Center . . . has been at the forefront of renewable energy initiative in Toronto" It is unclear here on how the Krotrick center serves the target of your article: a first step to standardization of small wind turbine. For instance, a standardized field measurement could provide typical roughness that can encounter a Small Wind Turbine such as typical roughness in cities . . . On what aspects the Kortrick measurement site is interesting for standardizing field data ?

P3 L62 Are you targeting standardization or on global warming problems ? The following sentence raises that doubt: "It is difficult to develop a small-scale diversified electrical generation strategy under a changing wind field caused by global warming"

A more focused analysis is needed for the reasons detailed below:

- Ashtine et al (2016) have performed a seasonal and long-term trends of the same NARR dataset. The present study only includes a non-linear filter on the NARR from the on-site calibrated power curve. We are therefore not surprised to have "trends in the electrical output that closely represent wind speed trends" (p8 L176).

- The on-site calibrated power curve corresponds to the winter season which, from the analysis performed by the author (p10 L223-224), is the greatest turbine output and thus overestimate the production for other seasons.

From my point of view, the originality of your work is the use of electrical power output from existing wind turbines. However, theses wind turbines are only used to calibrate the power curve using a much shorter dataset (6 month) than the reanalyzed database available (33years). We don't have variation of the power curve calibration during the seasonal cycle for instance. I guess you are targeting that objective and this work is a first approach with the available 6 month dataset ?

As the calibration power curve represents at least on season (winter), I would focus on that season for the analysis of the NARR dataset over 33years.

An additional bibliography is needed:

p3L54-56: The dataset used in this study exist since 1980. Also, I know that before choosing and installing a wind turbine site, this type of dataset is used (at least for a big wind turbine farm). Therefore I'm surprised of the poor literature you describe. By quickly looking for related articles on google I found these articles: "A review on the young history of the wind power short-term prediction", Renewable and Sustainable Energy Reviews Volume 12, Issue 6, August 2008, Pages 1725–1744 "Wind speed climatology and trends for Australia, 1975–2006: Capturing the stilling phenomenon and comparison with near-surface reanalysis output" Geophysical Research letters, VOL. 35, L20403, 2008

Their work are using power curves from manufacturers (from a very quick read of these studies ...). These studies (or other found by the authors) would help to enhance the need to use on-site power curved.

Minor issues:

p3 L76: define, at least once, the acronym NARR and maybe give an URL.

p4L85-86: Where are located the field measurements for the reanalysis ? Are they available at both height (10m and 30m) ? How many spatial points over the region targeted ?

p4 L89: please put the URL in a footnote for readability

p5 and later P6 L134: The power curve of the Skystream wind turbine is not necessary as no analysis based on this turbine is shown later.

p6 L135: " ... power curve demonstrated electrical output ..." I would attenuate the world "demonstrated" by using "predict" (or another chosen by the authors) as the work is based on a reanalyzed dataset (i.e. using modeling).

―――――――――――――――――

---

## Author Comment (AC2) · 12 Jul 2016

We would like to sincerely apologise for the delay in this response as we are still working out the appropriate timelines with the interactive discussion and presumed it to be closed once your review was received.

You raise many valid points, both on technical issues and when taking into consideration a the bigger picture that the article tries to disseminate. We see your review as an in depth and structured assessment of our article and thus will take every measure to address your points in our revised manuscript, making note of each change.

We hope you are pleased with your revised version when it is completed.

Best wishes